# Exploring the determinants of synergetic development of social organizations participating in home-based elderly care service: An SEM method

Qiuhu Shao[1☺], Jingfeng Yuan[1☺]*, Junwei Ma[1‡], Hongxing Ding[1‡], Wei Huang[2‡]

1 Department of Construction and Real Estate, School of Civil Engineering, Southeast University, Nanjing, Jiangsu Province, P. R. China, 2 School of Civil Engineering, Sanjiang University, Nanjing, Jiangsu Province, P. R. China

☺ These authors contributed equally to this work.
‡ These authors also contributed equally to this work.
* jingfeng-yuan@seu.edu.cn

**Data Availability Statement:** All relevant data are within the manuscript and its Supporting Information files.

## Abstract

The current aging service industry has problems in meeting the ever-increasing demand for the home-based elderly care service (HECS). Social organizations participating in HECS seems to be a promising way to address these problems but also raises new challenges, like uncoordinated cooperation among stakeholders, which could lead to low management efficiency and low service quality. However, Synergetic development can be promising to enhance the participation of social organizations and to improve social welfare. This study introduces a conceptual model to explore relationships between five determinants and synergetic development of social organizations participating in HECS. A structural equation model (SEM) based on questionnaire survey is used as a test methodology. The results indicated that stakeholder engagement plays a critical role in synergetic development in HECS, resource allocation can only be improved by institutional climate, and supervision capacity cannot facilitate information sharing. This study provides effective strategies and directions for the improvement of home-based elderly care services.

## Introduction

The ever-increasing proportion of the elderly has raised a big challenge to the elderly care service supply system. Take China for example, institution-based elderly care service is usually out of supply. As such, home-based elderly care service (HECS), defined as care service provided to the elderly who are living at home instead of at institution, has been advocated by the government to alleviate the institution's heavy care service supply pressure [1–3]. HECS provides the elderly with convenient and accessible care service [4, 5], but requires additional investment to support and promote such service provision. Then, social organizations (i.e., non-government organizations and non-profit organizations) are encouraged to cooperate with the government to co-provide the elderly care service since social organizations have

**Funding:** The research is supported by the National Natural Science Foundation of China (NO. 7207020615, NO.71671042; receiver: Jingfeng Yuan), National Social Science Foundation of China (NO. 19CGL065; receiver: Wei Huang). Both Yuan and Huang play a role in the study design, data collection and analysis, decision to publish, or preparation of the manuscript.

**Competing interests:** The authors have declared that no competing interests exist.

advantages of market operation, like powerful labour resources and advanced technological assets [6]. Typically, social organizations participating in HECS in the form of specialized pension institutions, community units, volunteers, and charitable organizations.

Despite advantages of social organization participating in HECS, new problems raise for the government. For example, how to encourage social organizations to engage in HECS? How to allocate various resources among different social organizations? How to ensure care service quality? HECS system is such a complex and relatively disordered system which have not achieved sustainable market operation. As such, synergetic development can be considered a promising approach to improving the HECS model. The idea of synergetic development originates from synergy theory which was first put forward by Hakon [7], who explained that a complex and disordered system can achieve order and sustainability by interaction among different subsystems. Existing studies concerning synergetic development mainly focus on public service, such as logistics transportation and ecological protection [8, 9], but pay little attention to the elderly care service. Partnering relationships among social organizations are studied to integrate different levels of health care services to provide low-cost care [10–12]. However, in the HECS, the stakeholders are complex, including the government, the community, social organizations and the elderly, meaning that coordination among different stakeholders for better care service is rather intractable.

From the perspective of synergy development, this paper defines synergetic development of social organizations participating in HECS as *coordinated development or joined-up development among different organizations to provide a potential solution for growing aging care service (i.e., better access to care service for the elderly, improved satisfaction and experience for the elderly, and high-efficient resource utilization for the whole society)* [13, 14]. Through analyzing the characteristics of HECS and social organizations, a theoretical model is proposed from the external and internal perspective. External influencing factors contain supervision capacity and institutional climate; internal factors include stakeholder engagement, resource allocation, and information sharing. By using Structural Engineering Model (SEM), this paper finds that stakeholder engagement plays a critical role in synergetic development in HECS, resource allocation can only be improved by institutional climate, and supervision capacity cannot facilitate information sharing. In addition, this paper highlights the importance of stakeholders in HECS and first examines the impact of policy and supervision on HECS in a qualitative way.

## Research method

SEM technique is a commonly used method to examine multivariate relationships, which consists of the measurement model test and the structural model test. The measurement model focuses on relationships between the observed variables and the latent variables. The structural model focuses on relationships between the latent variables and is used to test the fitting index of the proposed conceptual model. Confirmatory Factor Analysis (CFA) is applied to estimate the measurement model and assess the reliability and validity of construct measures. For good convergent validity of a model, Composite Reliability (CR) should be bigger than 0.7 and Average Variance Extracted (AVE) should be bigger than 0.5. According to [15], this paper applies six indices to examine the model fitness, as shown in Table 1. The logic of research design is shown as Fig 1.

### Research hypothesis

To achieve synergetic development of social organizations participating in HECS, this paper first explores determinants that influence the efficiency of these social organizations based on the theoretical framework of Ansell and Gash [16]. To achieve cooperation between

**Table 1. Indices and criteria for the good fitness of a model.**

| Indice | χ2/df | CFI | NFI | GFI | IFI | RMSEA |
|---|---|---|---|---|---|---|
| Criteria | ≤3 | >0.9 | >0.9 | >0.9 | >0.9 | <0.1 |

Note: CFI = Comparative Fit Index, NFI = Normed Fit Index, GFI = Goodness of Fit Index, IFI = Incremental Fit Index, RMSEA = Root Mean Square Error of Approximation (RMSEA).

government organizations and social organizations, Ansell and Gash showed that both institutional climate and facilitative leadership like supervision would influence collaborative processes, thus affecting collaborative outcomes. Information sharing, resource allocation, and stakeholder engagement are critical to facilitate collaboration among stakeholders. As such, this paper explores the following five variables that may affect the synergetic development of social organizations participating in HECS: resource allocation, information sharing, stakeholder engagement, institutional climate, and supervision capacity.

**Resource allocation.** HECS resources, including staff, funding, equipment, and so on, are significant for the elderly to attain efficient aging service [17]. If service supply is in short, aging service demand then cannot be well met. Similarly, the social aging service will achieve equity only if the resources are well allocated. Only if service resource is fully utilized can HECS be improved; Otherwise, some resources will be wasted while some elderly people with low incomes cannot obtain care service. Therefore, we put forward:

*H1. Resource allocation is positively related to synergetic development.*

**Information sharing.** Information sharing means demand information as well as supply information are circulating in the aging service market. Information sharing plays an important role in service delivery and corporation [18, 19]. Accurate information and efficient

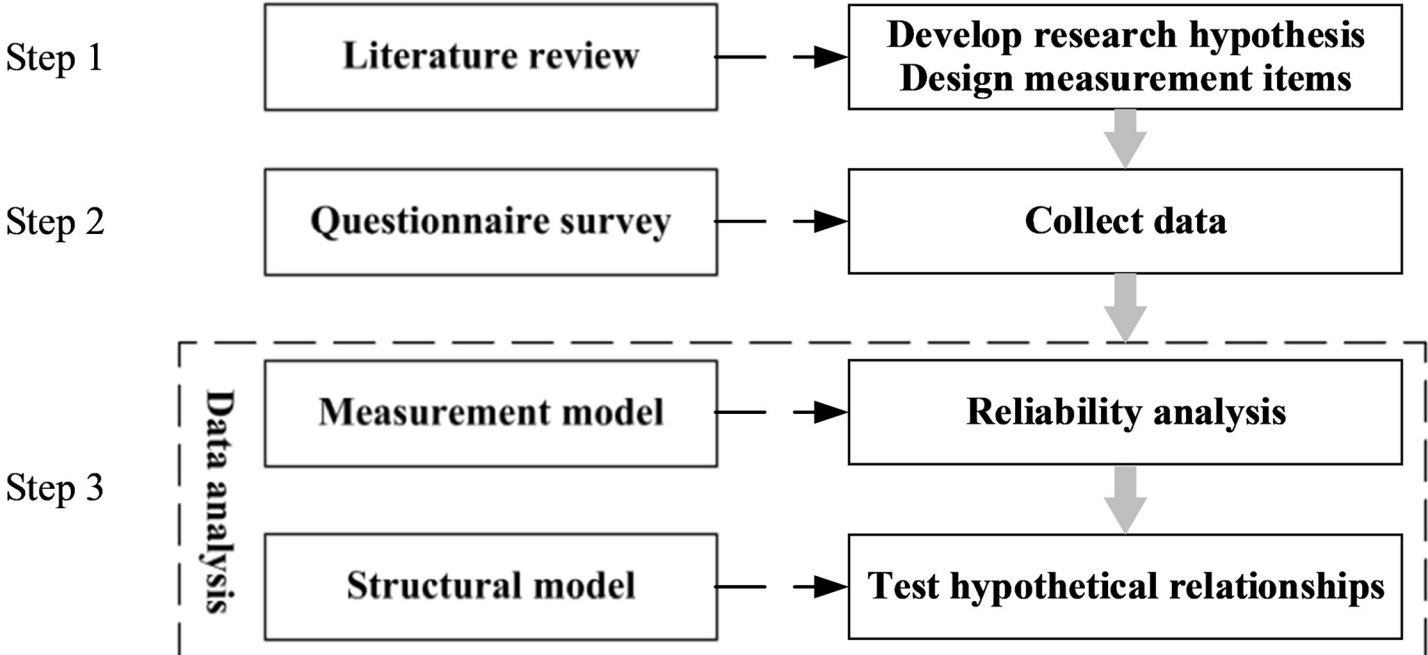

**Fig 1. The logic of research design.**

transmission help the service-providers informed well about service demand, like service preference, current location, and so on [20]. Then service resources can be delivered to the right place. Besides, information sharing in one service chain can also trigger decision adjustments in another service chain, which has a direct impact on resource integration [18]. To learn about the demand of the elderly clearly and to share this information among service-providers timely and accurately are beneficial to service delivery as well as conducive to improving the elders' sense of service contentment. Therefore, we put forward:

**H2a.** *Information sharing is positively related to resource allocation.*

**H2b.** *Information sharing is positively related to synergetic development.*

**Stakeholder engagement.** Stakeholder engagement refers to connection and corporation among different stakeholders in HECS. HECS involves four main stakeholders, namely, the government, the social organization, the community institution, and the elderly. In the traditional HECS model, the government is responsible for the total social aging service, from service production, service provision to service supervision, which has been proved to have low efficiency. Stakeholder engagement, however, could facilitate the resource and information to flow in the home-based care system [21]. By communication and interaction, resource allocation and information sharing are induced [22], thus promoting the sustainability of HECS. Therefore, we put forward:

**H3a.** *Stakeholder engagement is positively related to resource allocation.*

**H3b.** *Stakeholder engagement is positively related to information sharing.*

**H3c.** *Stakeholder engagement is positively related to synergetic development.*

**Institutional climate.** Institutional climate includes policy climate and market climate. The government has strong control over social resource due to its special status and has edges over making policy regarding service subsidies and nursery staffs to prosper the development of the aging service market [13]. Apart from the government, the market also plays a regulatory role in aging service delivery [23]. By setting the clear service process and the standard of quality, the market, to some extent, can overcome the problems of low service quality caused by information asymmetry. By clarifying the rights and responsibilities of all parties in the market, communication and cooperation can be well promoted. The institutional climate is not only a constraint but also a support to facilitate service delivery. Therefore, we put forward:

**H4a.** *Institutional climate is positively related to resource allocation.*

**H4b.** *Institutional climate is positively related to information sharing.*

**H4c.** *Institutional climate is positively related to stakeholder engagement.*

**H4d.** *Institutional climate is positively related to synergetic development.*

**Supervision capacity.** Social organizations participating in HECS needs the constant exchange of resource and information among various elements to maintain and facilitate the development of HECS. One main aspect of supervision capacity is to ensure the availability of aging services [24]. In the process of resource allocation, supervision capacity induces resources to flow in different areas, thus promoting the optimal combination of resources and improving the fairness and effectiveness of resource allocation. The supervision mechanism also stimulates the circulation of home-based care information throughout the service network. Through social supervision and media supervision, service providers will strengthen

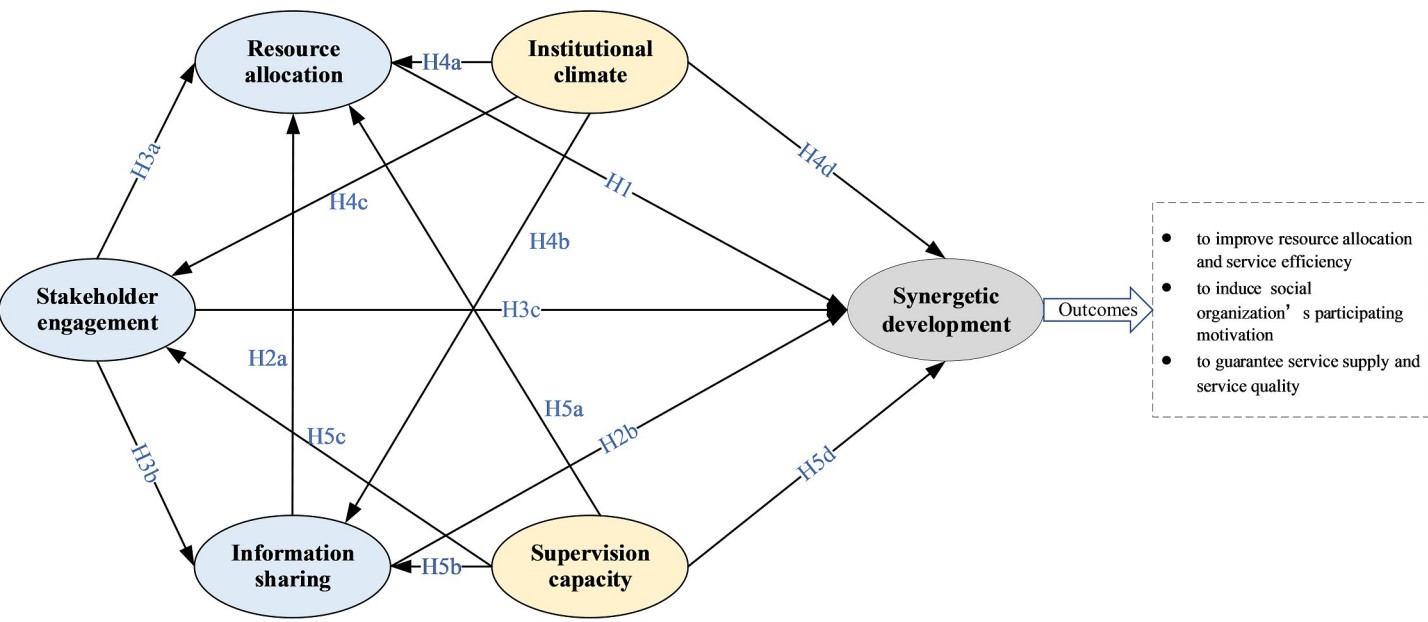

**Fig 2. The conceptual framework of synergetic development of social organizations participating in HECS.**

communication and interaction among all parties. For example, by constructing a service feedback mechanism, service providers can gain a deeper understanding of service requirements or problems in the service chain, thereby improving service quality and satisfaction. Therefore, we put forward:

***H5a.*** *Service supervision is positively related to resource allocation.*

***H5b.*** *Service supervision is positively related to information sharing.*

***H5c.*** *Service supervision is positively related to stakeholder engagement.*

***H5d.*** *Service supervision is positively related to synergetic development.*

The research hypothesis are summarized in Fig 2.

## Measurement items identification

Measurement items for each variable are then identified based on the revision of existing items in existing literature, as shown in Table 2. Currently, most of the funding and resources mainly come from the Chinese government, which is far from meeting the demand. Exploring diversified funding sources of funds (RA1) and attracting sufficient resources (talent, facilities, funds) (RA2) will improve the satisfaction of all stakeholders [17]. In fact, synergetic development could also be improved through the properly designed allocation of resources standards (RA3, RA4) which cannot only trigger enthusiasm but also result in lowered total costs and higher efficiency. Information sharing is fundamental to learn about the real demands as well as attract other's attention. In terms of social organizations participating in HECS, whether service providers know the demand of the elderly (IS2), and the accuracy (IS3) and efficiency (IS1) of information sharing will influence the perception of service providers regarding service demand [25, 26]. Besides, information technology increases the availability of each participant for information sharing (IS5) [27]. For the relationship among stakeholders, the close engagement provides opportunities for stakeholders to communicate with each other in

**Table 2. Measurement items of latent variables.**

| Latent variables | Observation variables | Measurement items |
|---|---|---|
| **Resource allocation** | RA1 | I think that stakeholders have different funding sources [17] |
| | RA2 | I think there are sufficient resources (like facilities and funds) for stakeholders to use [17] |
| | RA3 | I think that the allocation of resources (funds, facilities) helps to improve the efficiency of use [17] |
| | RA4 | I think the allocation of resources (talents, facilities, funds) is fair [17] |
| **Information sharing** | IS1 | I think the efficiency of information transmission (service demand, material allocation) among stakeholders is high [27] |
| | IS2 | I think the symmetry of information (service demand, material allocation) among stakeholders is high [25] |
| | IS3 | I think the accuracy of information (service demand, material allocation) transmission among stakeholders is high [26] |
| | IS4 | I think the sharing degree of information (service demand, material allocation) among stakeholders is high [25] |
| | IS5 | I think the application level of information technology is high [27] |
| **Stakeholder engagement** | SE1 | I think I have a close engagement with (other) social organizations [28] |
| | SE2 | I think I have a close engagement with (other) government departments [28] |
| | SE3 | I think I have a close engagement with (other) community institutions [28] |
| | SE4 | I think I have a close engagement with (other) the elderly [28] |
| **Institutional climate** | IC1 | I think that there are comprehensive subsidy policies to encourage the participation of stakeholders [29] |
| | IC2 | I think that there are comprehensive professionals training policies [30] |
| | IC3 | I think that entry and exit mechanism for social organizations is reasonable [31] |
| | IC4 | I think that power and responsibility mechanism for stakeholders is rational [32] |
| | IC5 | I think that there are clear standards to regulate the service process and quality [32] |
| **Supervision capability** | SC1 | I think there is a thorough performance evaluation system in social organizations [33] |
| | SC2 | I think community institutions can effectively supervise social organizations and government departments [34] |
| | SC3 | I think government departments can effectively fulfill their responsibilities (support, publicity, cooperation, and accountability) [34] |
| | SC4 | I think the public and the media can actively participate and supervise [34] |
| | SC5 | I think the opinions of the elderly can be effectively conveyed and implemented [35] |
| **Synergetic Development** | SD1 | I think synergetic development will improve the effectiveness of stakeholders' expenditure [36, 37] |
| | SD 2 | I think the popularization of synergetic development can attract more social organizations to participate [36, 37] |
| | SD 3 | I think synergetic development will improve service satisfaction [36, 37] |
| | SD 4 | I think synergetic development will improve social benefit [36, 37] |

HECS. Therefore, a close engagement among four main stakeholders (SE1, SE2, SE3, SE4) will improve the efficiency of cooperation and service supply [28]. Institutional climate includes subsidy policies (IC1), professionals training policies (IC2), clarified power and responsibility among stakeholders (IC3), reasonable entry and exit mechanisms (IC4), and clear standards (IC5) [29–32]. Supervision in HECS requires the engagement of each stakeholder. For the social organization, a standard evaluation mechanism needs to be established (SC1) [33]. For the community institution, it has the closest relationship with the elderly and needs to check the service quality carefully (SC2). The government also needs to be responsible for the aging service market (SC3) [34]. As for the elderly, they care about whether their demand is well met (SC5) [35]. In addition, social media also plays an important role in service supervision (SC4) [34]. Synergetic development of social organizations participating in HECS means high-efficient resource utilization (SD1), better access to the service market (SD2), and improved satisfaction and experience for the elderly (SD3), and improved social welfare (SD4) [36, 37].

## Data collection and analysis

Data was collected via the questionnaire. Before formal data collection, a pre-survey based on the questionnaire draft was conducted to ensure the consistency and accuracy of the questionnaires. With advice from 12 professionals, the questionnaire was revised, improved, and validated. Then the formal questionnaire was conducted for people and agencies related to elderly services in Nanjing by random sampling. The formal questionnaire consists of two parts. The first part is the background information of the respondents, including the gender, age, education, working experience, and job category. The second part is the question of measuring the latent variables using the Likert seven-point scale ranging from 1 ("strongly disagree") to 7 ("strongly agree") measurement. All procedures performed in this study involving human participation were with ethics approval from the Temporary Ethics Committee, School of Civil Engineering, Southeast University. All the participants were informed about the purpose of the study and their right to refuse participation or terminate their involvement during the study and informed consent were obtained. Participants of this study were requested to provide written informed consent before participation. All data analyzed anonymously in this study are available from the Nanjing Civil Affairs Bureau of China. SPSS 20.0 and AMOS 26.0 are further used to analyse the proposed conceptual model based on the principal of SEM.

We received a total of 249 responses, and 211 were valid. The effective response rate was 84.74%, which was deemed adequate for the data analysis. Among them, 80 questionnaires were from academic, 57 respondents were from the government department, 32 respondents were from service organizations, 42 respondents were from the elderly care association, which ensured the accuracy and scientificity of the results. More than 92% of the respondents have more than two years of elderly care work experience or research experience, which ensures the credibility and reliability of the questionnaire survey results. The background information of participants is presented in Table 3.

## Measurement model

Since all factor loadings should be above 0.7 to demonstrate that the indicator performs a satisfactory degree of reliability, two indicators (RA1 was 0.466, IT4 was 0.503) were deleted after the first CFA. After deleting RA1 and IT4, the new results of CFA suggested an acceptable fit between the measurement model and data set, where $\chi 2/df = 1.034$, CFI = 0.998, GFI = 0.901, NFI = 0.934, IFI = 0.998, and RMSEA = 0.013.

Table 4 shows that the values of Cronbach's Alpha ranged from 0.864 to 0.942, and the values of CR ranged from 0.871 to 0.943, which indicates good reliability for all of the constructs [38]. AVE ranged from 0.660 to 0.804, indicates adequate convergent validity [39].

## Structural model

The output results of the fitting index of the model ($\chi 2/df = 1.557$, CFI = 0.968, NFI = 0.915, IFI = 0.968, RMSEA = 0.052) suggested the extended model had an acceptable model fit. Institutional climate, stakeholder engagement, resource allocation, information sharing, and supervision capability explained 87% of the variance of synergetic development [40].

Fig 3 and Table 5 shows the hypotheses testing results. First, we assessed the relationships between determinants and synergetic development. We found that institution climate, supervision capacity, stakeholder connection, information sharing and resource allocation had a significant impact on synergetic development ($p<.05$). Then we assessed the relationships among the five determinants. External influencing factors (institution climate and supervision capacity) were firstly tested. Institution climate had a significant impact on stakeholder connection ($p<.001$), information sharing ($p<.05$), and resource allocation ($p<.001$). Supervision capacity

Table 3. Background information of participants.

| Demographics | Frequencies | Percentages (%) |
|---|---|---|
| **Gender** | | |
| Male | 114 | 54 |
| Female | 97 | 46 |
| **Age** | | |
| <30 | 38 | 18 |
| 30-40 | 59 | 28 |
| 40-50 | 70 | 33 |
| >50 | 44 | 21 |
| **Education level** | | |
| High school | 23 | 11 |
| bachelor degree | 38 | 18 |
| Graduate degree | 70 | 33 |
| Ph.D. degree | 80 | 38 |
| **Job category** | | |
| government department | 57 | 27 |
| Experts and scholars | 80 | 38 |
| Service organizations | 32 | 15 |
| Elderly care association | 42 | 20 |
| **Work experience (years)** | | |
| <2 | 15 | 7 |
| 2-5 | 59 | 28 |
| 5-8 | 84 | 40 |
| >8 | 53 | 25 |

also had a significant impact on stakeholder engagement (p<.001) and resource allocation (p<.001). However, there was no significant direct relationship between supervision capacity and information sharing (p>.05), which failed to support H5b. The relationship among internal influencing factors (stakeholder engagement, information sharing, and resource allocation) were tested lastly. Stakeholder engagement had a significant impact on information sharing (p<.001) and had no significant direct impact on resource allocation (p>.05), which supported H3b, not H3a. Information sharing had no significant direct impact on resource allocation (p>.05), which failed to support H2a.

## Discussion

### Direct effect of the internal variables on synergetic development

According to the results of SEM, resource allocation (H1, standardized coefficient = 0.207), information sharing (H2b, standardized coefficient = 0.256), and stakeholder engagement (H3c, standardized coefficient = 0.284) have positive influence on synergetic development. The allocation of resources, including human resources, equipment resources, and capital resources, plays a pivotal role in the elderly care industry. In this process, as Lotfi et al. have mentioned, information sharing enables stakeholders to obtain all kinds of knowledge [41]. Also, the various elements, including institution design and service supervision, are organically linked through information flow, so as to eliminate the isolation of stakeholders and realize the multiple combinations of obligations, responsibilities, and benefits. Resources, information and stakeholders are the key elements for the synergetic development of all industries [42, 43], and SEM results show that they are no exception in the elderly care industry.

**Table 4. Results of confirmatory factors analysis.**

| Constructs | Items | Factor loadings | Cronbach's alpha | CR | AVE |
|---|---|---|---|---|---|
| Institutional Climate | IC1 | 0.800 | 0.906 | 0.907 | 0.660 |
| | IC2 | 0.822 | | | |
| | IC3 | 0.822 | | | |
| | IC4 | 0.819 | | | |
| | IC5 | 0.798 | | | |
| Stakeholder Engagement | SE1 | 0.846 | 0.898 | 0.898 | 0.687 |
| | SE2 | 0.833 | | | |
| | SE3 | 0.825 | | | |
| | SE4 | 0.812 | | | |
| Resource Allocation | RA2 | 0.826 | 0.864 | 0.871 | 0.691 |
| | RA3 | 0.842 | | | |
| | RA4 | 0.826 | | | |
| Information Sharing | IS1 | 0.813 | 0.893 | 0.894 | 0.678 |
| | IS2 | 0.821 | | | |
| | IS3 | 0.835 | | | |
| | IS5 | 0.824 | | | |
| Supervision Capability | SC1 | 0.817 | 0.904 | 0.907 | 0.660 |
| | SC2 | 0.825 | | | |
| | SC3 | 0.824 | | | |
| | SC4 | 0.820 | | | |
| | SC5 | 0.775 | | | |
| Synergetic Development | SD1 | 0.900 | 0.942 | 0.943 | 0.804 |
| | SD2 | 0.892 | | | |
| | SD3 | 0.905 | | | |
| | SD4 | 0.889 | | | |

## Direct effect of the external variables on synergetic development

The two hypotheses between the external variables and synergetic development are supported by SEM. Institutional climate and supervision capacity are respectively associated with synergetic development with standardized coefficient values of 0.147 and 0.235. Communication barriers have been a major obstacle in the supervision of long-term elderly care recently. Since supervision engagement is a supporting variable that can effectively develop synergy, promoted social media should be used for supervision. Such a positive hypothesis can be identified in the previous literature [44, 45]. In addition, institutional climate directly influences synergetic development. Although the relationship between institutional climate and synergetic development is partially mediated by affecting stakeholder connection, the path of institutional climate and synergetic development is still significant.

## Indirect effect of the external variables on synergetic development via internal variables

Institutional climate is the fundament to improve the synergetic development of social organizations participating in HECS. First, institutional climate is positively associated with resource allocation (H4a) with standardized coefficients of 0.362, which verifies that institutional forces of legitimacy are not iron cages. Fair policies and competitive markets tend to integrate idle social resources, thus to improve resource utilization efficiency and improve the elders' satisfaction. The point is consistent with the findings of Garg etc., which revealed policies can

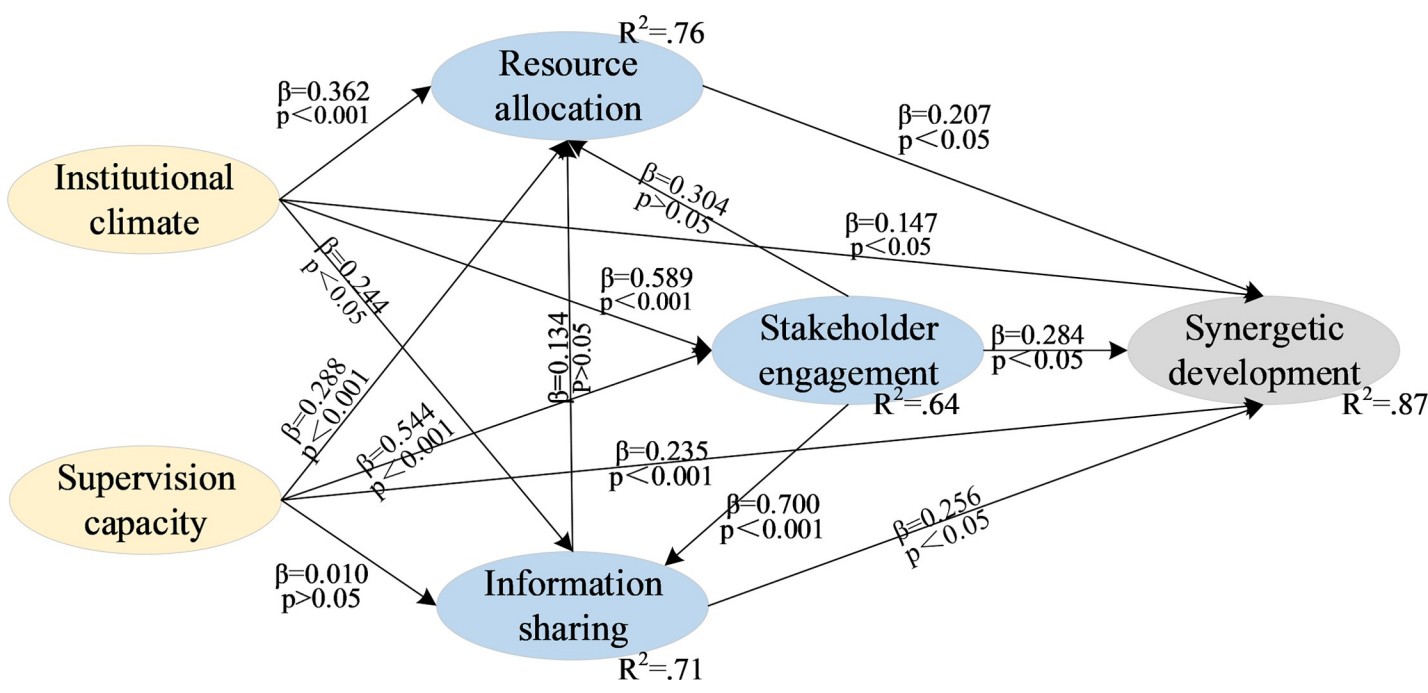

**Fig 3. The results of the proposed model.**

ensure optimal allocation of scarce healthcare resources [46]. Second, the relationship between institutional climate and information sharing is verified (H4b, standardized coefficient = 0.244). It is easy to understand that institutional guarantee will provide perfect information exchange

**Table 5. Results of the structural equation modeling.**

| Paths | Coefficients | t-values | Hypothesis | Result |
|---|---|---|---|---|
| RA→SD | 0.207 | 2.358* | H1 | **Supported** |
| IS→RA | 0.134 | 0.917 | H2a | **Not Supported** |
| IS→SD | 0.256 | 2.346* | H2b | **Supported** |
| SE→RA | 0.304 | 1.903 | H3a | **Not Supported** |
| SE→IS | 0.700 | 5.739*** | H3b | **Supported** |
| SE→SD | 0.284 | 2.340* | H3c | **Supported** |
| IC→RA | 0.362 | 4.003*** | H4a | **Supported** |
| IC→IS | 0.244 | 2.757** | H4b | **Supported** |
| IC→SE | 0.589 | 8.369*** | H4c | **Supported** |
| IC→SD | 0.147 | 2.051* | H4d | **Supported** |
| SC→RA | 0.288 | 3.544*** | H5a | **Supported** |
| SC→IS | 0.01 | 0.120 | H5b | **Not Supported** |
| SC→SE | 0.544 | 8.024*** | H5c | **Supported** |
| SC→SD | 0.235 | 3.667*** | H5d | **Supported** |

Note

*$p < 0.05$

**$p < 0.01$

***$p < 0.001$, SD = Synergetic Development, RA = Resource Allocation, IS = I nformation Sharing, SE = Stakeholder Engagement, IC = Institutional Climate,
SC = Supervision Capacity.

channels for stakeholders in the elderly care industry, which provides support across the continuum of care. Third, institutional climate positively relates to stakeholder engagement (H4c) with the highest standardized coefficients of 0.589. Previous literatures point out that fair elderly care policies and competitive environment are the key factors to induce stakeholder to engage and clarify the rights and responsibilities [47]. Combined with the positive association between institutional climate and resource allocation, institutional climate can significantly promote stakeholder engagement. In this case, stakeholders can give full play to their advantages to form a good industrial order.

Supervision capacity also did well in promoting synergy, though one hypothesis is not supported. First, supervision capacity remains significantly associated with resource allocation (H5a) with standardized coefficient values of 0.288. The efficient supervision strategy of governments not only minimizes waste of resources but also helps in exploiting idle resources available in the HECS system. Resources integrated during such supervision could greatly improve the efficiency of elderly care service. Likewise, supervision is highly beneficial for stakeholder engagement (H5c, standardized coefficient = 0.544). In the future, multi-stakeholder supervision in the elderly care industry could focus on jointly working and knowledge sharing between government and social organizations. However, supervision has not a positive influence on information sharing (H5b), which conflicts with the literature [48]. On the one hand, supervision standards are not comprehensive enough to deal with information accessed from diversified channels. For example, social media supervision standard is lacking, which prevents access to high-quality feedback from the elderly. On the other hand, even though information has been collected by elderly care platforms, it does not mean that the ways of information sharing are efficient.

## Interaction among the internal variables

Stakeholder engagement is positively related to information sharing (H3b) with standardized coefficient values of 0.700. Corporation among stakeholders can integrate professional work in elderly care services, thus breaking down information exchange barriers and improving the operation efficiency of HECS.

However, information sharing is not positively related to resource allocation (H2a) with standardized coefficient values of 0.134. From a macro perspective, the shared information, such as favorable policy or market demand, will prompt more social organizations to enter the industry. However, as the resources of the whole industry are limited at present, social organizations will plunder these resources, leading to unfair resource allocation. Such an explanation can be seen from the previous literatures, which mainly focus on early elderly care practices in developed countries [3, 49]. From a micro perspective, social organizations in Chinese are in resource shortage at present, which causes such information sharing is of no significance. For example, shared information that most social organizations lack professional talent has not significantly improved talent resources allocation, and the industry still faces the problem of a talent shortage for a long time. Only by resolving the severe resources shortage in the industry can we fundamentally address the problem of information sharing.

In addition, stakeholder engagement does not have significant influence on resource allocation (H3a). This result may be related to the insignificance of the above hypotheses. Stakeholder engagement can promote information sharing, but information sharing cannot promote resource allocation, so stakeholder engagement cannot improve resource allocation. As one of the most intractable problems in elderly care industry, the question that how to allocate resource should be determined by institutions and supervision, rather than stakeholders who are self-interested.

## Implications

Collaboration, communication, and coordination are of great importance to the synergetic development of social organizations participating in HECS. To enhance collaboration, stakeholder engagement is fundamental [21]. Meanwhile, the authorities play a guidance role in the development of HECS [50], who are able to launch several incentive policies, such as subsidy mechanisms and staff introduction strategies, to attract service providers including social organizations and community institutions to participate in the aging industry. It is also a must for governments to clarify the duty of each party and the standard of service delivery via policy and rules. As for service providers, they could form an alliance to deal with some difficulties, such as resource shortage and communication barriers. They could also build a platform to strengthen the bonds among different service suppliers and service demanders. As a whole, it is crucial for service providers to innovate the model of service provision with the help of technology [50]. As far as the elderly are concerned, their awareness of using aging care services should be aroused by the public, and they should change their traditional views towards aging and adapt to HECS. Institutes for senior citizens can be constructed to improve the elders' living skills and consumer awareness. Moreover, young citizens should also be included in the elderly care service by some activities like time bank, to cultivate their consumption awareness regarding aging service in advance.

To improve communication, information sharing, institutional guarantee, and service supervision are equally important. With the growing aging population, a demand database is helpful to store and deal with numerous demand information. By artificial intelligence and big data, changes in demand can be captured timely and accurately, then personalized service could be delivered to each old people. The institutional guarantee means that the government could discover some aging problems via policies and remind service providers of some business opportunities. The service feedback mechanisms can also help service providers improve the service quality as well as keep pace with the changes in the service demand. Apart from learning about service demand, a compatible and public service system is helpful to reduce repetitive work and improve service efficiency [51].

Finally, to achieve coordination, public-private-partnership are suggested in HECS [52]. By cooperation between governments and social organizations, resource shortage, information barriers, and some other problems can be alleviated. By delegating social organization to establish digital systems and technological infrastructures, governments not only provide the social organization with business chances but also manage the aging industry in a good manner.

## Conclusion

This study investigated the determinants of the synergetic development of social organizations participating in HECS. These determinants can be divided into two external variables and three internal variables. Through the literature review, this study explored the indicators of these determinants. The data on the determinants and indicators were collected from elderly agencies and research agencies in Nanjing, China. A conceptual model was proposed to explain the relationships between synergetic development and these determinants. An SEM was conducted to test the proposed conceptual model and identify the major determinants affecting the synergetic development of social organizations participating in HECS.

The results of the SEM indicate all the determinants have a positive direct impact on the synergetic development of social organizations participating in HECS. The nine hypothetical relationships among external variables and internal variables were tested by the SEM method. Most hypothetical relationships were positively supported except H2a, H3a, H5b. The results indicated the direction of stakeholder engagement on the synergetic development were

identified to be more important than other determinants. In addition, five determinants can indirectly influence the synergetic development by affecting other determinants. Furthermore, several useful suggestions can be drawn from the research findings from the perspective of government, social organizations, community, and elderly people.

Although this study can effectively enhance the participation of social organizations and the efficiency of the elderly care industry, the results of this study should be interpreted in light of the following limitations. First, more determinants should be investigated by the actual situation. Second, more indirect influence on synergetic development should be investigated in the future. In addition, more data and information should be stratified for analysis based on government, social organizations, community, and the elderly to avoid some deviations.

## Supporting information

**S1 Data. Survey data.**
(XLSX)

## Author Contributions

**Conceptualization:** Qiuhu Shao, Jingfeng Yuan, Hongxing Ding, Wei Huang.

**Data curation:** Qiuhu Shao, Junwei Ma.

**Formal analysis:** Qiuhu Shao, Jingfeng Yuan, Hongxing Ding.

**Funding acquisition:** Jingfeng Yuan, Wei Huang.

**Investigation:** Qiuhu Shao, Junwei Ma, Hongxing Ding, Wei Huang.

**Methodology:** Junwei Ma, Wei Huang.

**Project administration:** Jingfeng Yuan.

**Resources:** Jingfeng Yuan.

**Supervision:** Jingfeng Yuan.

**Validation:** Jingfeng Yuan, Junwei Ma.

**Visualization:** Junwei Ma.

**Writing – original draft:** Qiuhu Shao, Junwei Ma, Hongxing Ding, Wei Huang.

**Writing – review & editing:** Qiuhu Shao, Jingfeng Yuan, Junwei Ma, Hongxing Ding.

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
