## [Decision Letter · Decision Letter 0]

18 Nov 2020

PONE-D-20-20452

Exploring the Determinants of Synergetic Development of Social Organizations Participating in Home-based Elderly Care Service: An SEM Method

PLOS ONE

Dear Dr. Yuan,

Thank you for submitting your manuscript to PLOS ONE. After careful consideration, we feel that it has merit but does not fully meet PLOS ONE’s publication criteria as it currently stands. Therefore, we invite you to submit a revised version of the manuscript that addresses the points raised during the review process.

 Please see comments below

We look forward to receiving your revised manuscript.

Kind regards,

Dejan Dragan, PhD

Academic Editor

PLOS ONE

Journal Requirements:

Additional Editor Comments:

Dear Author,

according to the review of both reviewers, the major revision is required.

AE DD

Reviewers' comments:

Reviewer's Responses to Questions

**Comments to the Author**

1. Is the manuscript technically sound, and do the data support the conclusions?

Reviewer #1: Yes

Reviewer #2: Yes

2. Has the statistical analysis been performed appropriately and rigorously? 

Reviewer #1: Yes

Reviewer #2: Yes

3. Have the authors made all data underlying the findings in their manuscript fully available?

Reviewer #1: Yes

Reviewer #2: Yes

4. Is the manuscript presented in an intelligible fashion and written in standard English?

Reviewer #1: No

Reviewer #2: Yes

5. Review Comments to the Author

Reviewer #1: Dear author

Thanks for your valuable article. But the article has its drawbacks.

Article segmentation is not well done. The performance is cluttered and confuses the reader. The volume of content is very high and could have been more key.Comments are given in the text of the article.

Reviewer #2: Review Report (PONE-D-20-20452)

Ref: (PONE-D-20-20452)

Title: Exploring the Determinants of Synergetic Development of Social Organizations Participating in Home-based Elderly Care Service: An SEM Method

Journal: PLOS ONE

I've read with attention this paper and I believed that this study could be of interest to many readers and helpful for health policy makers. synergy is one of the main components in high level of systems theory, and it can increase synergistic thoughts and actions in workplace environment. This study explores determinant factors on synergetic management in home-based elderly care service using a literature review, and then examines the relationship between variables using structural equation model. However, I believed that this study has many flaws and need a major revision.

1. Abstract:

The abstract of the paper is not clearly written. In abstract, background, objective, methods, results and conclusion must be summarized. But, in this abstract we can not find the purpose of the study and conclusion. Also, the methodology of this research in abstract is not clearly. I recommend that the keywords revised based on MeSH database.

2. Introduction:

Introduction starts well and we get acquainted with changes in population structure and changes in family structure. These changes create a demand for home-based elderly care services. But it does not continue as well. Introduction does not have a proper hierarchy. It is better that you review the previous work similar to your research in introduction. Also, many of content in literature review section is repeated. I suggest that literature review can be merge into introduction. I fined two heading title “Social organizations participating in HECS” in literature review.

However, the biggest problem with the introduction is that you have included factors influencing synergetic management in the introduction.

The other problem is that conceptual model and hypothetical relationships should be transferred to the method and results section.

First, you have to explain how choose the determinant factors. What search strategy were used and which databases did you search?

3. Methods:

SEM is a powerful technique to test and evaluate multivariate relationships.

The main problem in this study is related to methods. methodology of the paper needs more elaboration and clarity. I recommended that rewrite the methods of your manuscript and explain the methodology step-by-step. How you selected the variables affecting on synergetic management?

How you selected the measurement items and design the questionnaire?

Also, please provide the steps of SEM model in your research clearly.

Table 2 and sociodemographic characteristics of participants should be presented in the results section.

4. Results:

Many contents of results should have been presented in the method section (such as line 325-327, 336-337 and 340). Overall results are good.

5. Discussion:

The discussion is the repeat of results and justifications have not been given on how and why current study results are different or alike to the past studies. I recommend that to support the discussion based on the reported results with the past researches, so that it clearly justify present research.

6. Limitation:

It is suggest that provide the limitation of your work.

Overall this study is a good paper with useful results but there are many flaws specially in methods and discussion. If the authors of the paper disagree with the comments/suggestions, then provide suitable justifications.

6. PLOS authors have the option to publish the peer review history of their article (what does this mean?). If published, this will include your full peer review and any attached files.

Reviewer #1: No

Reviewer #2: **Yes: **Javad Javan-Noughabi

---

## [Author Response · Author response to Decision Letter 0]

16 Dec 2020

Journal Requirements:

Reply: The format of the whole manuscript has been revised according to the requirement of the journal PLOS ONE.

2. In your Data Availability statement, you have not specified where the minimal data set underlying the results described in your manuscript can be found. PLOS defines a study's minimal data set as the underlying data used to reach the conclusions drawn in the manuscript and any additional data required to replicate the reported study findings in their entirety. All PLOS journals require that the minimal data set be made fully available.

Upon re-submitting your revised manuscript, please upload your study’s minimal underlying data set as either Supporting Information files or to a stable, public repository and include the relevant URLs, DOIs, or accession numbers within your revised cover letter.

Reply: Underlying data has been uploaded to the system in the file named Supporting Information files.

3. Please include captions for your Supporting Information files at the end of your manuscript, and update any in-text citations to match accordingly.

Reply: Thanks for your comments. We have revised accordingly.

Reviewer 1#:

I've read with attention this paper and I believed that this study could be of interest to many readers and helpful for health policy makers. synergy is one of the main components in high level of systems theory, and it can increase synergistic thoughts and actions in workplace environment. This study explores determinant factors on synergetic management in home-based elderly care service using a literature review, and then examines the relationship between variables using structural equation model. However, I believed that this study has many flaws and need a major revision.

1. Abstract:

The abstract of the paper is not clearly written. In abstract, background, objective, methods, results and conclusion must be summarized. But, in this abstract we can not find the purpose of the study and conclusion. Also, the methodology of this research in abstract is not clearly. I recommend that the keywords revised based on MeSH database.

Reply: Abstract has been rewritten and Keywords have been deleted according to the requirement of the journal.

2. Introduction:

Introduction starts well and we get acquainted with changes in population structure and changes in family structure. These changes create a demand for home-based elderly care services. But it does not continue as well. Introduction does not have a proper hierarchy. It is better that you review the previous work similar to your research in introduction. Also, many of content in literature review section is repeated. I suggest that literature review can be merge into introduction. I fined two heading title “Social organizations participating in HECS” in literature review. 

However, the biggest problem with the introduction is that you have included factors influencing synergetic management in the introduction.

The other problem is that conceptual model and hypothetical relationships should be transferred to the method and results section.

First, you have to explain how choose the determinant factors. What search strategy were used and which databases did you search?

Reply: Introduction has been reorganized by simplifying previous Introduction and merging Literature review. Literature review section has been merged into Introduction in the revised version. Line 44-49 introduce the advent of home-based care service (HECS) and the participance of social organizations in HECS. Line 51-52 present the problems raised by social organizations participating HECS based on existing literature. Line 52-57 explain the reason to apply synergetic development in HECS. Line 57-62 show existing studies on synergetic development and partnership among social organizations, as well as the research gap. And the final paragraph of the Introduction is the summary and highlight of the whole paper’s work.

The analysis of factors influencing synergetic management have been removed to Section Research method, please refer to Line 91-99, and the details of five influencing factors analysis are in the subsection Research hypothesis.

Conceptual model and hypothesis have been removed into the Method section.

The determinants are based on the framework of Ansell and Gash. To achieve cooperation between government organizations and social organizations, Ansell and Gash showed that both institutional climate and facilitative leadership like supervision would influence collaborative processes, thus affecting collaborative outcomes. Information sharing, resource allocation, and stakeholder engagement are critical to facilitate collaboration among stakeholders. As such, this paper explores the following five variables that may affect the synergetic development of social organizations participating in HECS: resource allocation, information sharing, stakeholder engagement, institutional climate, and supervision capacity. (Line 91-99)

3. Methods:

SEM is a powerful technique to test and evaluate multivariate relationships. 

The main problem in this study is related to methods. methodology of the paper needs more elaboration and clarity. I recommended that rewrite the methods of your manuscript and explain the methodology step-by-step. How you selected the variables affecting on synergetic management?

How you selected the measurement items and design the questionnaire?

Also, please provide the steps of SEM model in your research clearly.

Table 2 and sociodemographic characteristics of participants should be presented in the results section. 

Reply: Research method section has been rewritten. At the beginning of Section Research method, the step of SEM technique is stated, as shown in Fig 1 (Line 88-89). 

In the Research hypothesis, we explained that the variables are chosen based on the framework of Ansell and Gash. To achieve cooperation between government organizations and social organizations, Ansell and Gash showed that both institutional climate and facilitative leadership like supervision would influence collaborative processes, thus affecting collaborative outcomes. Information sharing, resource allocation, and stakeholder engagement are critical to facilitate collaboration among stakeholders. As such, this paper explores the following five variables that may affect the synergetic development of social organizations participating in HECS: resource allocation, information sharing, stakeholder engagement, institutional climate, and supervision capacity. (Line 91-99)

In the Measurement items identification, measurement items are identified based on the previous literature, as shown in Table 2.

The questionnaire is used to collect data. Before formal data collection, a pre-survey based on the questionnaire draft was conducted to ensure the consistency and accuracy of the questionnaires. With advice from 12 professionals, the questionnaire was revised, improved, and validated. Then the formal questionnaire was conducted for people and agencies related to elderly services in Nanjing by means of random sampling. The questionnaire consists of two parts. The first part is the background information of the respondents, including the gender, age, education, working experience, and job category. The second part is the question of measuring the latent variables using the Likert seven-point scale ranging from 1 (“strongly disagree”) to 7 (“strongly agree”) measurement. (Line 190-197)

The analysis of sociodemographic characteristics of participants has been removed to the section Data collection and analysis.

4. Results: 

Many contents of results should have been presented in the method section (such as line 325-327, 336-337 and 340). Overall results are good.

Reply: Thanks for your comments. Line 325-327, 336-337 and 340 in the previous manuscript regarding the introduction of measurement model and structural model have been removed to the section Research method, please refer to Line 76-87.

5. Discussion:

The discussion is the repeat of results and justifications have not been given on how and why current study results are different or alike to the past studies. I recommend that to support the discussion based on the reported results with the past researches, so that it clearly justify present research. 

Reply: First, the discussion was reorganized, and was divided into direct effect of the internal variables on synergetic development, direct effect of the external variables on synergetic development, indirect effects of the external variables on synergetic development via internal variables, and interaction among the internal variables according to the relationship types in the SEM model. Secondly, combining Table 5 in the results, standardized coefficients were added to the discussion of hypotheses, which made them more consistent with the results. Third, previous studies have been applied in the discussion, please see Line 250-251, 253-254, 261-262, 272-273, 278-280, 290-291. 

6. Limitation:

It is suggest that provide the limitation of your work.

Reply: Limitation has been provided in Line 363-368.

Overall this study is a good paper with useful results but there are many flaws specially in methods and discussion. If the authors of the paper disagree with the comments/suggestions, then provide suitable justifications.

Reply: Thanks for your comments. We have revised the whole manuscript seriously.

 

Reviewer 2#:

Comment: Abbreviations should be given in a footnote.

Reply: The format of the whole manuscript has been revised according to the requirement of the journal PLOS ONE.

Comment: What is the reason for segmentation at the end of the introduction? The layout of an article is different from a dissertation and it is not necessary to mention the last paragraph of the introduction.

Reply: The format of the whole manuscript has been revised according to the requirement of the journal PLOS ONE. Introduction has been rewritten and simplified. And the layout has been deleted.

Comment: The volume of content provided for an article is very large. It is better that the mentioned sections contain more concise and key points.

Reply: Literature review has been merged into Introduction. Key ideas are about synergetic development and social organizations participating in public service, which have been merged into Introduction (Line 44-62).

Comment: Where is the reference to this sentence? (“In addition, the relevant items were identified.”) The type of study is not specified correctly. The execution method does not have the necessary continuity. How is the systematic part of the plan done? According to which keywords? What studies have been used? Explain clearly.

Reply: Research method section has been rewritten. At the beginning of Section Research method, the step of SEM technique is stated, as shown in Fig 1 (Line 88-89). 

In the Research hypothesis, we explained that the variables used in our conceptual model are chosen based on the framework of Ansell and Gash. To achieve cooperation between government organizations and social organizations, Ansell and Gash showed that both institutional climate and facilitative leadership like supervision would influence collaborative processes, thus affecting collaborative outcomes. Information sharing, resource allocation, and stakeholder engagement are critical to facilitate collaboration among stakeholders. As such, this paper explores the following five variables that may affect the synergetic development of social organizations participating in HECS: resource allocation, information sharing, stakeholder engagement, institutional climate, and supervision capacity. (Line 91-99)

In the Measurement items identification, measurement items are identified based on the previous literature, as shown in Table 2.

Comment: Data collection should be in the results section, not in the method.

Reply: Data collection has been removed to the section Data collection and analysis.

---

## [Editor Report · Decision Letter 1]

18 Dec 2020

Exploring the d eterminants of synergetic development of social organizations participating in home-based elderly care service: An SEM method

PONE-D-20-20452R1

Dear Authors,

We’re pleased to inform you that your manuscript has been judged scientifically suitable for publication and will be formally accepted for publication once it meets all outstanding technical requirements.

Kind regards,

Dejan Dragan, PhD

Academic Editor

PLOS ONE

Additional Editor Comments (optional):

All comments were appropriately followed in the paper. Accordingly, the paper deserves an opportunity to be accepted. AE DD
---

## [Editor Report · Acceptance letter]

22 Dec 2020

PONE-D-20-20452R1 

Exploring the determinants of synergetic development of social organizations participating in home-based elderly care service: An SEM method 

Dear Dr. Yuan:

I'm pleased to inform you that your manuscript has been deemed suitable for publication in PLOS ONE. Congratulations! Your manuscript is now with our production department. 

Kind regards, 

on behalf of

Dr. Dejan Dragan 

Academic Editor

PLOS ONE